# EventSR-Zero: Training-free Event Video Super-Resolution with Diffusion Priors

## Abstract

Event-to-Video (E2V) methods aim to reconstruct intensity frames from events, bridging the gap between event-based and image-based vision. However, existing E2V approaches often fail to recover fine structures, leading to reconstructions with artifacts and degraded quality. To address this, we explore the task of Event-to-Video Super-Resolution (EVSR), which aims to reconstruct high-resolution video from low-resolution events. We present EventSR-Zero, a training-free framework that exploits the high temporal resolution of event cameras to recover fine-grained details from low-resolution events and uses them to guide a diffusion-based Video Super-Resolution (VSR) model in generating high-quality super-resolved videos of the underlying scene. Our approach incorporates two key components: (1) an Implicit Contrast Refinement (ICR) module that robustly extracts sub-pixel scene details from low-resolution events, and (2) a Reconditioning Guidance (RG) module that reliably steers the diffusion VSR process using the high-resolution event signal from ICR. Extensive experiments demonstrate that EventSR-Zero achieves state-of-the-art performance, surpassing existing event-based super-resolution methods. We will release our source code upon acceptance.

## 1 Introduction

Event cameras represent a significant advancement in vision technology, addressing the limitations of traditional frame-based cameras in high-speed and high-dynamic-range scenarios. Unlike conventional cameras that capture entire frames at fixed intervals, event cameras operate asynchronously, detecting per-pixel brightness changes with microsecond precision. Their sub-millisecond resolution, wide dynamic range, and sparse, memory-efficient output enable high-speed, low-latency vision while reducing storage and computational demands. Event cameras are crucial in robotics and autonomous systems for real-time navigation and obstacle detection Falanga et al. (2020); Forrai et al. (2023); Huang et al. (2024). They also enhance object recognition Zubić et al. (2024); Gehrig & Scaramuzza (2023); Li et al. (2021); Mitrokhin et al. (2019);

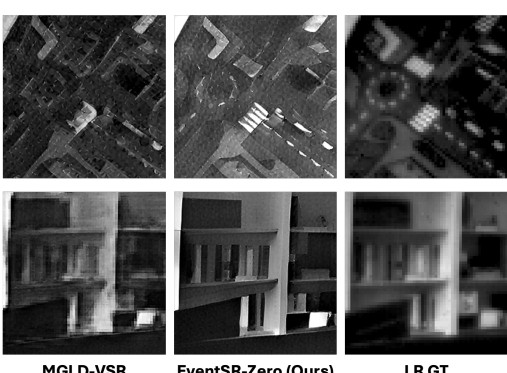

MGLD-VSR     EventSR-Zero (Ours)     LR GT

Figure 1: Our EventSR-Zero guides a diffusion VSR model with high-resolution structures recovered from events to generate high-quality SR frames from low-resolution events.

Scheerlinck et al. (2018) and surveillance Bi et al. (2019); Freeman et al. (2024); Verma et al. (2024), particularly in low-light conditions. In AR/VR, they enable precise gesture tracking and interaction Plizzari et al. (2022); Gao et al. (2023; 2024).

Event-to-video (E2V) reconstruction is crucial for bridging the gap between event-based and conventional vision systems, enabling broader adoption of event cameras in existing applications. Since event cameras output sparse, asynchronous data that encode only brightness changes, they lack absolute intensity information which makes direct interpretation challenging. Reconstructing videos from events restores intensity frames, allowing seamless integration with traditional computer vi-

sion pipelines that rely on frame-based processing. E2V reconstruction also enhances applications such as high-speed video capture, where traditional cameras struggle with motion blur, and low-light imaging, where event cameras excel due to their high dynamic range.

Research on deep-learning based E2V reconstruction Rebecq et al. (2019); Scheerlinck et al. (2020); Stoffregen et al. (2020a); Paredes-Valles & de Croon (2021); Cadena et al. (2021); Weng et al. (2021); Mostafavi et al. (2020); Wang et al. (2020); Ercan et al. (2024) aims to convert raw event streams into video frames by training over paired event-image datasets. This transformation enables more intuitive and accessible representations of the rich temporal and structural information contained within sparse, unstructured event data. However, despite significant progress, existing E2V techniques still struggle to recover fine details, particularly in complex scenes or high-speed scenarios, where the inherent sparsity of event data and the absence of absolute intensity values lead to artifact-laden or low-quality reconstructions.

Building on the limitations of conventional Event-to-Video (E2V) methods, Event-to-Video Super-Resolution (EVSR) introduces a significant advancement by enhancing the spatial resolution of reconstructed frames. Unlike traditional cameras that capture purely spatial information, event cameras record a spatiotemporal stream with microsecond temporal precision. This high temporal resolution enables the capture of subtle displacements that provide complementary spatial cues over time, thereby allowing inference of details beyond the native resolution of the sensor. Leveraging this property, EVSR can reconstruct high-frequency scene structures and generate super-resolved intensity images directly from event data.

By incorporating sub-pixel event alignment strategies with deep image priors, EVSR can significantly improve image sharpness and edge definition, making event-based vision more practical for high-precision applications such as autonomous navigation, medical imaging, and high-speed video capture, where fine visual details are crucial. Moreover, achieving high-resolution reconstructions from events enables better compatibility with existing high-resolution vision models, expanding the applicability of event cameras in mainstream computer vision tasks. The development of robust EVSR techniques can unlock the full potential of event cameras, enabling not just fast and low-latency vision but also high-quality, detailed imagery suitable for a wide range of real-world applications.

A direct approach is to train a high-resolution (HR) event-to-video model to learn the SR prior in event space. However, real-world HR event datasets do not exist, and furthermore existing strategies to simulate HR event datasets are still encumbered by a sim-to-real domain gap. As a result, EVSR remains a complex, ongoing challenge in event-based vision research.

In this work, we propose EventSR-Zero, a training-free approach that robustly recovers fine-grained details from low-resolution events by exploiting the high temporal resolution of event cameras and uses these details to guide a diffusion-based VSR model to generate highly detailed frames from low-resolution event streams.

Our method comprises two key components:

The **Implicit Contrast Refinement (ICR)** module enables recovery of high-frequency sub-pixel details from LR event data. It formulates a high-resolution contrast maximization (CMax) space, regularized by a frequency-constrained lightweight MLP that serves as an implicit function to mitigate the high propensity of event collapse in HR space.

2) The **Reconditioning Guidance (RG)** module is a novel diffusion guidance strategy that steers the diffusion trajectory through controlled adjustments to the conditioning image. At each step, RG derives frame estimates from intermediate latents and aligns their flow-directed spatial gradients with the HR details recovered by ICR, thereby transferring fine event structures into the conditioning. This ensures that the diffusion process generates frames whose structural features remain consistent with the high-resolution event details provided by ICR.

Our method does not utilize task-specific training data, thus alleviating the requirement for high-resolution event-image datasets. Qualitative and quantitative results show that our EventSR-Zero outperforms existing baselines to achieve effective event video SR. Our **main contributions** are as follows:

- We introduce EventSR-Zero, a training-free method that achieves high quality super-resolved intensity images from LR events.
- We propose Implicit Contrast Refinement (ICR) that mitigates event collapse in HR space to reliably recover sub-pixel level details from high-temporal resolution events.
- We propose Reconditioning Guidance (RG), a novel diffusion guidance technique that produces event-guided, super-resolved intensity frames while maintaining consistency towards underlying HR scene structures.

## 2 RELATED WORKS

### 2.1 EVENT TO VIDEO SUPER-RESOLUTION (EVSR)

Reconstructing intensity images from events is a key topic in event-based vision, with various methods offering different assumptions and processing techniques. Early approaches Cook et al. (2011); Kim et al. (2014); Agrawal et al. (2005); Kim et al. (2016); Barua et al. (2016); Aharon et al. (2006); Bardow et al. (2016); Munda et al. (2018); Scheerlinck et al. (2018) relied on simplified assumptions in constrained camera motion or brightness constancy to reconstruct intensity frames. More recently, deep learning methods Rebecq et al. (2019); Scheerlinck et al. (2020); Stoffregen et al. (2020a); Paredes-Vallés & de Croon (2021); Cadena et al. (2021); Weng et al. (2021); Mostafavi et al. (2020); Wang et al. (2020); Ercan et al. (2024) have achieved state of the art results in intensity image reconstruction. These methods adopt voxelized event grids to encode sparse events, and typically operate a recurrent network to capture long-range context from past event segments. These methods are trained over low-resolution event-image datasets and are effective at video reconstruction at low-resolution. Across existing methods, Hyper-E2VID Ercan et al. (2024) currently achieves the highest reconstruction fidelity through the combination of a recurrent event voxel encoding architecture and a dynamic filter generation hypernetwork.

In the area of Event to Video Super-Resolution (EVSR), super-resolved intensity frames are produced from pure event streams. Mostafavi et al. (2020) parses events as stacked event images and implements an optical flow estimator, a feature rectification network, a recurrent SR network, and a mixer network to reconstruct HR frames. The modules are trained end-to-end over a ESIM Rebecq et al. (2018) simulated HR dataset to achieve intensity image SR. Wang et al. (2020) proposes three sequential networks for reconstruction, restoration and SR that are trained end-to-end over a simulated EventSR dataset that is also generated from ESIM Rebecq et al. (2018). Duan et al. (2021) proposes a display-camera system for HR event data collection, which is used to train a U-Net based framework to estimate HR spatiotemporal event point clouds.

Similarly, our work seeks to produce super-resolved event-based intensity images solely from event streams. Unlike existing approaches, our EventSR-Zero does not require synthetic HR event datasets. In our work, we use Hyper-E2VID Ercan et al. (2024) to generate the initial LR frames for event-guided VSR. We also adopt the formulation of Zhang et al. (2023) to provide event-based guidance on the diffusion trajectory by aligning the spatial gradients of the intermediate image estimates with the Image of Warped Events (IWE) that contain HR details captured from ICR.

### 2.2 DIFFUSION-BASED VIDEO SUPER-RESOLUTION

Diffusion models transform a sample from a noise distribution to a target data distribution using a fixed forward process and a learned reverse denoising process. The reverse process is learned by a network $\phi$ that is trained to denoise a noised latent $\mathbf{z}_t$ by predicting its noise component $\epsilon_\phi(\mathbf{z}_t, y, t)$ conditioned upon $y$ (text/image) and timestep $t$. Diffusion-based Video Super-Resolution (VSR) models take a sequence of LR images as conditioning for the diffusion network $\phi$ and adds SR details to the generated image across multiple diffusion steps. The reverse process adds detail to the generated image by iteratively denoising a noise sample $\mathbf{z}_T$ into a sample $\mathbf{z}_0$ from the data distribution across a diffuison trajectory.

Recently, diffusion-based single-image SR models Rombach et al. (2022); Wang et al. (2024) have been adapted for Video Super-Resolution (VSR) Yang et al. (2024); Zhou et al. (2024), incorporating fine-tuning and specific constraints to manage temporal coherence and motion consistency across frames. These adaptations allow diffusion-based VSR models to set new benchmarks in video super-

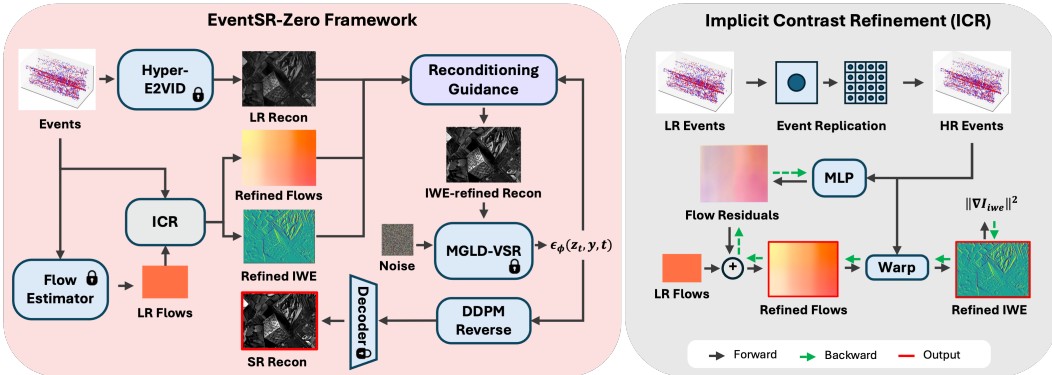

Figure 2: (Left) EventSR-Zero enhances LR reconstructions from HyperE2VID using a diffusion VSR model (MGLD-VSR) guided by Reconditioning Guidance (RG), which aligns latent reconstructions with HR IWEs and flows from our Implicit Contrast Refinement (ICR) module. (Right) ICR replicates events over subpixels and encodes HR coordinates with frequency-controlled positional embeddings, passing them through a lightweight MLP to estimate smooth HR flow residuals.

resolution quality. In this work, we utilize a frozen MGLD-VSR Yang et al. (2024) model guided by motion-compensated events to achieve high-quality event-based video super-resolution.

### 2.3 CONTRAST MAXIMIZATION

Contrast maximization (CMax) is a technique in event-based vision that optimizes flow estimates for motion compensation, improving event alignment and enhancing scene edges. It has been applied to motion estimation Gallego et al. (2018) and intensity reconstruction Zhang et al. (2023), where contrast-maximized stacks sharpen visual features. However, CMax is prone to event collapse Shiba et al. (2022a), where events aggregate into patterns that achieve high contrast but obscure true structure. To mitigate this, recent works impose hierarchical grids that interpolate pixel flows from coarser estimates Shiba et al. (2022b), or derive flow via nearest-neighbor associations to nonlinear motion trajectories Friedhelm Hamann (2024). Others reduce collapse risk by restricting motion model degrees of freedom, e.g., assuming rotational or 6-DOF egomotion Zhang et al. (2023). While these strategies improve stability, they oversimplify real-world scene dynamics and cannot capture complex non-rigid motions. In contrast, EventSR-Zero departs from prior work by applying CMax in a high-resolution *refinement* stage rather than at the native event resolution. Specifically, EventSR-Zero introduces a high-resolution optimization space that simulates events from a finer-resolution sensor, allowing convergence to more detailed scene structures. To address the heightened risk of event collapse in this enlarged space, it employs a frequency-constrained implicit function regularizer that stabilizes the optimization.

## 3 OUR METHOD

Our EventSR-Zero framework integrates an event-to-video reconstruction model Hyper-E2VID Ercan et al. (2024) with a diffusion-based video super-resolution (VSR) model MGLD-VSR Yang et al. (2024) to create a backbone for event video super-resolution. Hyper-E2VID initially reconstructs low-resolution (LR) frames from raw event streams, which are then super-resolved into high-resolution (HR) frames by MGLD-VSR. However, the VSR model cannot differentiate between artifacts and true features in the LR input. Without proper guidance, the ill-posed nature of super-resolution where multiple SR solutions can correspond to a single LR input causes this pipeline to risk generating details that deviate from the true scene structure.

To address the above-mentioned issues, we introduce a training-free event-informed guidance that tweaks the diffusion trajectory to help produce details that are aligned with HR structural features derived from the event stream. Our approach includes two main components: 1) An **Implicit Contrast Refinement (ICR)** module to recover high-resolution details from events; 2) A **Reconditioning Guidance (RG)** module to provide precise diffusion trajectory adjustments for the VSR model.

Fig. 2 (left) illustrates the EventSR-Zero framework. Raw events are processed by Hyper-E2VID Ercan et al. (2024) and an event-based flow estimator (Shiba et al. (2022b)) to produce initial LR reconstructions and LR flows, respectively. Our ICR then upscales the LR flows to HR flows, producing refined IWEs to assist in diffusion guidance. The MGLD-VSR model Yang et al. (2024) takes the LR reconstructions from Hyper-E2VID as conditioning inputs for the diffusion-based super-resolution process. At each diffusion step, the Reconstruction Guidance (RG) module extracts intermediate reconstructions from the diffusion latents and aligns their spatial gradients with the refined IWEs, which embed high-frequency scene details. These enriched reconstructions are iteratively recombined into the conditioning images, guiding subsequent diffusion steps to generate super-resolved frames with improved fidelity to fine-grained structures captured by the event data.

### 3.1 IMPLICIT CONTRAST REFINEMENT

Fig. 2 (right) outlines Implicit Contrast Refinement (ICR), which begins with event replication: each pixel in the LR event sensor is subdivided to simulate a higher-resolution sensor. For $4\times$ upscaling, every LR pixel corresponds to 16 subpixels in HR space. This is achieved by replicating each LR event across the equivalent patch of HR pixels, with each HR event given its respective spatial offset. This enables the refinement process to consolidate details in a higher spatial resolution. This event replication crucial for initializing the ICR process is formulated as:

$$e^{lr}_{x,y,t,p} = \{e^{hr}_{kx,ky,t,p}, ..., e^{hr}_{kx+(k-1),ky+(k-1),t,p}\} \tag{1}$$

where $x, y, t, p$ is the location, time and polarity, and $k = 4$ is the scaling factor for $4\times$ SR. We run Contrast Maximization (CMax) as a refinement step in HR space to recover HR flow solutions that reveal finer scene structures from the LR event stream. This is done by finding HR flow estimates that maximize the alignment of corresponding events. Specifically, a warp function $\mathbf{f}(\cdot)$ maps each event in a set $\mathcal{E} = \{e^i_{x,y,t,p}\}^{N_e}_{i=1}$ to a new position:

$$e_{x',y',t',p} = \mathbf{f}(e_{x,y,t,p}, t'). \tag{2}$$

The warped events are accumulated onto an Image of Warped Events (IWE) defined as:

$$I_{iwe}(\mathbf{p}; \mathbf{f}) \doteq \sum_{i=1}^{N_e} p_k \delta(\mathbf{p} - \mathbf{f}(\mathbf{p}'_i)), \tag{3}$$

where each pixel $\mathbf{p}$ accumulates the polarities of events that fall within its spatial region. The contrast of the IWE is measured by the gradient magnitude:

$$\|\nabla I_{iwe}\|^2 = \frac{1}{N_p} \sum_{i,j} \left( I_x^2(\mathbf{p}_{i,j}) + I_y^2(\mathbf{p}_{i,j}) \right), \tag{4}$$

where $\nabla I = (I_x, I_y)$ is the gradient of $I_{iwe}$, $N_p$ is the number of pixels, $I_x \equiv \frac{\partial I}{\partial x}$, and its magnitude is measured by the $L2$ norm.

To stabilize convergence, we optimize zero-initialized residuals on top of the bilinearly upsampled LR flows, constraining deviations from the initial estimates. The refined HR flow is obtained as

$$\mathbf{f}'_{hr} = \mathbf{U}(\mathbf{f}_{lr}) + \Delta\mathbf{f}_\theta, \tag{5}$$

where $\Delta\mathbf{f}_\theta$ denotes the residual correction.

Combining equations 3, 4, 5, ICR executes per-scene optimization:

$$\underset{\Delta\mathbf{f}_\theta}{\arg\min} \|\nabla I_{iwe}(\mathbf{p}; \mathbf{f}'_{hr})\|^2. \tag{6}$$

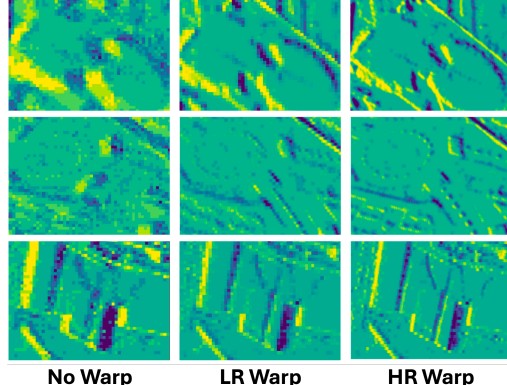

**No Warp**     **LR Warp**     **HR Warp**

Figure 3: Comparison of IWEs under different warping regimes. No warp (left), LR CMax (middle), HR CMax via ICR (right).

Fig. 3 demonstrates how CMax in HR space is able to recover fine-grained details. The increased spatial capacity from HR sensor simulation allows CMax to recover finer edge structures that are encoded in motion that are beyond the native sensor resolution.

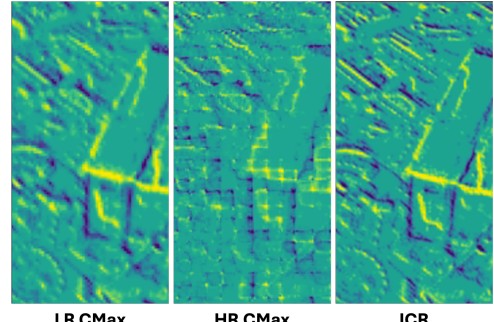

Figure 5: Our Reconditioning Guidance (RG) adds training-free control of the diffusion trajectory by carefully altering the conditioning image over diffusion steps. The 1-step reconstructions from intermediate latents are refined via the alignment of spatial gradients against the IWE, and the refined image is transferred to the conditioning image by exponential moving average.

However, the risk of event collapse in CMax increases substantially as the flow parameters under optimization acquire additional degrees of freedom. In the high-resolution domain, where the number of flow parameters grows by a factor of 16, collapse becomes inevitable. Fig. 4 (middle) illustrates this degeneracy: when CMax is applied directly in HR space, unconstrained flows concentrate events at regular spatial intervals, producing grid-like patterns in the IWE. Although these patterns yield high contrast scores, they show severely distorted scene structure and erase genuine details.

Our ICR is designed to mitigate event collapse in HR space via a local flow smoothing constraint that exploits the implicit smoothness of MLP to learn a continuous flow function:

$$u_{x,y}, v_{x,y} = G_\theta(\gamma_J(x), \gamma_J(y)), \qquad (7)$$

where $u, v$ are residual flow estimates at pixel $(x, y)$, $G_\theta$ is the MLP, and $\gamma_J(\cdot)$ represents a positional encoding function with $J$ frequency bases. We employ sinusoidal positional encodings Vaswani et al. (2023) on the query coordinates, retaining only low-frequency components to increase the similarity of positional encodings between local coordinates. We use $J = 4$ in our ICR module and we also apply average pooling for final smoothing of flow residuals. Fig. 4 illustrates the benefits of ICR. ICR enables contrast maximization to be achieved in HR space by

Figure 4: CMax in HR space degenerates with event collapse (middle). ICR uses frequency-constrained positional encodings in a lightweight MLP to regularize against degenerate HR flow solutions.

adopting a residual flow optimization process and providing implicit flow regularization to avoid event collapse while supporting fine-grained detail recovery from event data.

## 3.2 RECONDITIONING GUIDANCE

Reconditioning Guidance uses refined IWEs from ICR to guide the diffusion trajectory of a frozen MGLD-VSR Yang et al. (2024) model. Fig. 5 illustrates the RG process. At each diffusion step, noise predictions $\epsilon_\phi(\mathbf{z}_t, y, t)$ form one-step reconstructions $\hat{\mathbf{z}}_0$ which are decoded to image estimates $\hat{\mathbf{x}}_0$. We then transfer fine-grained details from ICR by aligning $\hat{\mathbf{x}}_0$ to the refined IWE, yielding aligned frames $\bar{\mathbf{x}}_0$. $\bar{\mathbf{x}}_0$ is then blended with conditioning image $\mathbf{x}_{\text{cond}}$ by exponential moving average (EMA) to update the conditioning $y$ for the next diffusion step.

The alignment of $\hat{\mathbf{x}}_0$ to the refined IWE requires the matching of visual features across different modalities (intensity image vs IWE). To establish this link, we draw on Zhang et al. (2023), which formulates intensity image reconstruction as a linear inverse problem. Here, the IWE is approximated as the spatial derivative of the ground-truth intensity frame $I_{gt}$:

$$D_{x,y}I_{gt} \approx I_{iwe}(x, y), \qquad (8)$$

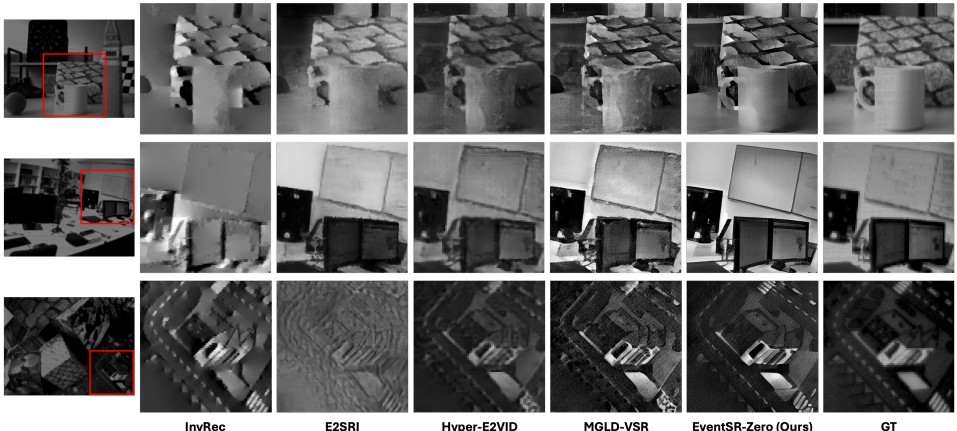

InvRec     E2SRI     Hyper-E2VID     MGLD-VSR     EventSR-Zero (Ours)     GT

Figure 6: Qualitative Results on ECD Dataset.

where $D_{x,y}$ denotes a finite difference operator. To align an image $\ell$ with the IWE, we minimize:

$$\min_{\ell} ||D_{x,y}\ell - I_{iwe}||^2 + \lambda\mathcal{R}(\ell), \tag{9}$$

where $\ell$ is the image being optimized, and $\lambda\mathcal{R}(\ell)$ is a regularization term. By enforcing flow-directed spatial gradient matching between $\hat{\mathbf{x}}_0$ and the refined IWE, fine-grained structures recovered by ICR are effectively transferred into the image estimate.

RG is applied at every diffusion step, where $\hat{\mathbf{x}}_0$ is refined for $M = 100$ iterations using Eq. (9) with an L1 loss to limit deviations and a Total Variation loss Rudin et al. (1992) to suppress artifacts, producing detail-enhanced estimates $\bar{\mathbf{x}}_0$.

The refined $\bar{\mathbf{x}}_0$ is then used as conditioning for the next diffusion step. While we experimented with self-guidance Epstein et al. (2023), its impact on SR outputs was negligible, likely because SR models rely heavily on the conditioning image, making them less responsive to intermediate latent adjustments. To address this limitation, our RG approach introduces a new control mechanism that directly adjusts the conditioning image during diffusion.

Our initial strategy was to fully replace the conditioning image with IWE-aligned reconstructions by setting $\mathbf{x}_t^{\text{cond}} = \bar{\mathbf{x}}_0$. However, this approach caused errors from earlier diffusion steps to be reinjected into the conditioning, creating a feedback loop that amplified artifacts. To overcome this, we introduce an Exponential Moving Average (EMA) update that gradually incorporates IWE-aligned details while maintaining stability from past conditioning images. EMA effectively balances guidance and robustness by preventing abrupt changes that propagate errors across steps. Formally,

$$\mathbf{x}_{t-1}^{\text{cond}} = \eta\bar{\mathbf{x}}_0 + (1 - \eta)\mathbf{x}_t^{\text{cond}}, \tag{10}$$

where $\eta$ determines the trade-off between event guidance and stability. Large $\eta$ values risk overemphasizing $\bar{\mathbf{x}}_0$ and amplifying artifacts, while small values underutilize IWE information.

Empirically, we found $\eta = 0.05$ to be optimal for effective IWE guidance. The combination of spatial gradient alignment and EMA blending ensures that the diffusion process remains aligned with event-based observations over diffusion steps, guiding the VSR model to produce details that are structurally consistent with the scene.

## 4 EXPERIMENTS

In this section, we present both qualitative and quantitative results to highlight the effectiveness of the proposed EventSR-Zero model for $4\times$ event video upscaling.

### 4.1 EXPERIMENT SETTINGS

We benchmark EventSR-Zero against three baseline models: 1) **InvRec.** The linear inverse reconstruction technique from Zhang et al. (2023). 2) **E2SRI.** The event-video super-resolution model

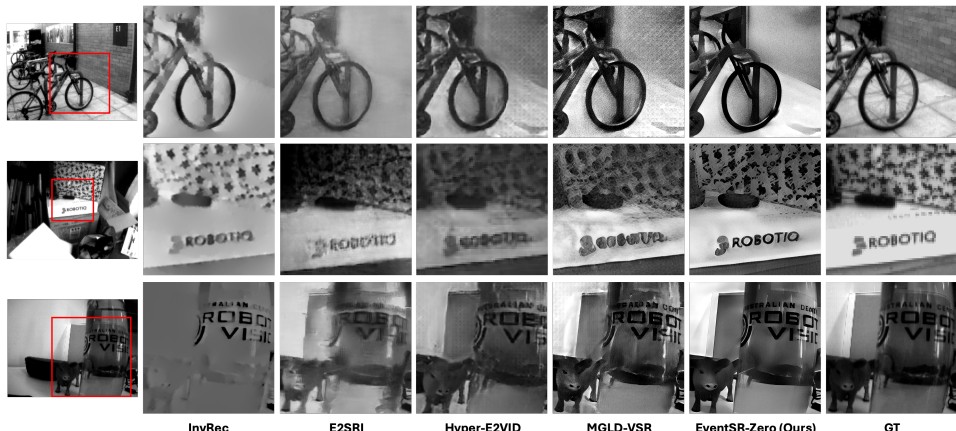

InvRec     E2SRI     Hyper-E2VID     MGLD-VSR     EventSR-Zero (Ours)     GT

Figure 7: Qualitative Results on HQF Dataset.

introduced in Mostafavi et al. (2020). 3) **MGLD-VSR.** Our own baseline model with low-resolution frames generated by Hyper-E2VID Ercan et al. (2024) and upscaled using MGLD-VSR Yang et al. (2024).

## 4.2 DATASET AND METRICS

For evaluation, we use sequences from two real-world datasets: the Event Camera Dataset (ECD) Mueggler et al. (2017) and the High-Quality Frames (HQF) Dataset Stoffregen et al. (2020b).

**Event Camera Dataset (ECD).** ECD was captured using a DAVIS240C sensor to provide both events and frames at a resolution of 240 × 180, with ground truth frames recorded at 22 Hz. Following Rebecq et al. (2019), we select seven short sequences of static office scenes with 6-DOF camera motion.

**High-Quality Frames Dataset (HQF).** HQF consists of indoor and outdoor scenes with diverse motion patterns, captured with a DAVIS240C sensor at 240 × 180 resolution. Ground truth frames are recorded at 22.5 Hz and are carefully selected to ensure minimal motion blur.

**Metrics.** We evaluate our method using three full-reference metrics: 1) Mean Squared Error (MSE); 2) Structural Similarity (SSIM) Wang et al. (2004); 3) Learning Perceptual Image Patch Similarity (LPIPS) Zhang et al. (2018) to assess alignment with the true scenes structure. For the full-reference metrics, generated SR frames are downsampled to the original resolution for comparison with ground truth frames.

Since high-resolution ground truth frames are unavailable, we also apply two no-reference metrics: Naturalness Image Quality Evaluator (NIQE) Mittal et al. (2013) and BRISQUE Mittal et al. (2012) to assess the perceptual quality of SR images at high resolution. During evaluation, we perform histogram equalization on all reconstructions and ground truth images.

## 4.3 QUALITATIVE RESULTS

The qualitative comparisons are presented in Fig. 6 and Fig. 7. GT refers to the LR ground truth frames. Our EventSR-Zero consistently reconstructs sharper edges and finer details compared to other methods. For example, our EventSR-Zero resolves the edges of the whiteboard more accurately in Fig. 6 row 2. This improvement is driven by our Reconditioning Guidance (RG), which effectively guides the diffusion process using high-resolution structures recovered by Implicit Contrast Refinement (ICR). This combination of fine-grained structure guidance and robust image priors of MGLD-VSR enables EventSR-Zero to produce clear and structured reconstructions of the whiteboard. The closeness of results from our EventSR-Zero to the structures in the ground truth (despite being low-resolution) highlights the effectiveness of ICR and RG in preserving alignment with scene structures derived from event data.

Table 1: Quantitative results of existing methods and our EventSR-Zero on sequences from ECD and HQF datasets. Best score is in bold and runner-up score is underlined.

| | ECD | | | | | HQF | | | | |
|---|---|---|---|---|---|---|---|---|---|---|
| | MSE ↓ | SSIM ↑ | LPIPS ↓ | NIQE ↓ | BRISQUE ↓ | MSE ↓ | SSIM ↑ | LPIPS ↓ | NIQE ↓ | BRISQUE ↓ |
| InvRec Zhang et al. (2023) | 0.060 | 0.422 | 0.245 | 8.572 | 52.161 | 0.084 | 0.316 | 0.246 | 7.414 | 46.359 |
| E2SRI Mostafavi et al. (2020) | 0.066 | 0.419 | 0.226 | 6.236 | 28.131 | 0.073 | 0.313 | 0.231 | 6.379 | 33.004 |
| MGLD-VSR Yang et al. (2024) | 0.061 | 0.378 | 0.242 | 3.449 | 25.494 | 0.080 | 0.302 | 0.240 | 3.793 | 25.869 |
| Ground Truth | - | - | - | 9.312 | 68.742 | - | - | - | 9.075 | 65.852 |
| EventSR-Zero (Ours) | **0.052** | **0.451** | **0.217** | **3.276** | **20.123** | **0.067** | **0.331** | **0.203** | **3.505** | **21.307** |

The MGLD-VSR baseline in Fig. 6 column 4 further illustrates the impact of our ICR and RG modules. This baseline applies MGLD-VSR Yang et al. (2024) directly to Hyper-E2VID Ercan et al. (2024) outputs, and therefore is effectively our EventSR-Zero model without ICR and RG. We observe that on its own, MGLD-VSR tends to super-resolve existing artifacts and errors from Hyper-E2VID such as grid-like patterns as shown in Fig. 6 columns 3 and 4, without distinguishing these artifacts from true structures. In contrast, our EventSR-Zero with ICR and RG significantly sharpens edges and reduces artifacts to yield more refined SR outputs.

## 4.4 QUANTITATIVE RESULTS

Quantitative results are shown in Tab. 1. Our EventSR-Zero outperforms competing baselines in MSE, SSIM, and LPIPS. Together, RG and ICR demonstrate complementary strengths in enhancing the alignment between super-resolved outputs and true scene structures. On no-reference metrics, our EventSR-Zero significantly outperforms other methods in NIQE Mittal et al. (2013) and BRISQUE Mittal et al. (2012) scores, indicating a substantial improvement in perceptual quality. The SR frames produced by EventSR-Zero exhibit enhanced sharpness and detail, contributing to a more realistic appearance.

**Discussion.** Since HR ground truth images are unavailable, full-reference evaluation necessitates downsampling the super-resolved (SR) outputs to match the lower-resolution ground truth for comparison. Although these low-resolution ground truth images retain the scene's structural information, their poor visual quality makes them an imperfect reference for evaluating downsampled SR frames. This limitation is evident in their extremely high NIQE/BRISQUE scores (Table 1, Row 5).

## 4.5 ABLATIONS

We ablate the performance contributions of RG, EMA and ICR in Tab. 2. The results show that both ICR and RG noticeably enhance reconstruction accuracy and visual quality across all metrics. ICR first recovers high-resolution scene structures

Table 2: Ablations of ICR and RG on HQF dataset.

| RG | EMA | ICR | MSE ↓ | SSIM ↑ | LPIPS ↓ | NIQE ↓ | BRISQUE ↓ |
|---|---|---|---|---|---|---|---|
| | | | 0.080 | 0.302 | 0.240 | 3.793 | 25.869 |
| ✓ | ✓ | | 0.072 | 0.324 | 0.215 | 3.652 | 23.070 |
| ✓ | | ✓ | 0.093 | 0.266 | 0.291 | 3.591 | 21.847 |
| ✓ | ✓ | ✓ | **0.067** | **0.331** | **0.203** | **3.505** | **21.307** |

to offer more precise event guidance, followed by RG which leverages diffusion SR priors to produce high-quality event-guided details. Furthermore, EMA is essential for RG to produce accurate details with minimal hallucination. Without EMA (3rd row), the reference-based metrics (MSE/SSIM/LPIPS) worsen significantly due to the emergence of severe hallucinative artifacts.

## 5 CONCLUSION

In summary, EventSR-Zero introduces a training-free framework for event-to-video super-resolution that directly exploits the temporal richness of event data. The Implicit Contrast Refinement (ICR) module formulates a high-resolution contrast maximization space, regularized by a frequency-constrained implicit function, to recover high-frequency sub-pixel details while mitigating event collapse. Complementing this, the Reconditioning Guidance (RG) module steers the diffusion process by aligning the spatial gradients of intermediate latents with ICR outputs, ensuring that reconstructed frames preserve high-resolution scene structures. Together, ICR and RG enable structurally consistent, high-quality reconstructions from low-resolution events without requiring high-resolution training data, demonstrating the potential of training-free event-guided diffusion to bridge the gap between sparse event streams and high-resolution video in real-world vision applications.

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
