## A  HYPERPARAMETER ABLATIONS

Fig. 1 evaluates the effect of varying the number of optimization iterations in both ICR and RG. We observe that performance steadily improves with additional iterations up to around 250 for ICR and 100 for RG, after which the

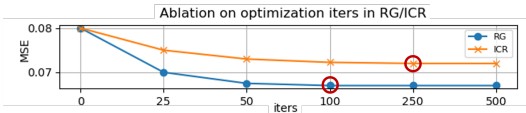

Figure 1: Ablations on ICR & RG iterations.

gains saturate. Beyond these thresholds, further iterations do not yield noticeable improvements in detail recovery or structural fidelity, but they increase computational cost. This demonstrates that our chosen settings of 250 iterations for ICR and 100 for RG strike a good balance between accuracy and efficiency, ensuring convergence without unnecessary overhead.

Fig. 2 illustrates the effect of varying $\eta$ values in Reconditioning Guidance (RG), which controls the guidance strength. High $\eta$ values ($\eta \geq 0.5$) strengthen the EMA update but introduce hallucinated artifacts as feedback loops cause detail errors to accumulate in the conditioning chain, causing deviations from the ground truth structure. Conversely, very low values ($\eta \leq 0.01$)

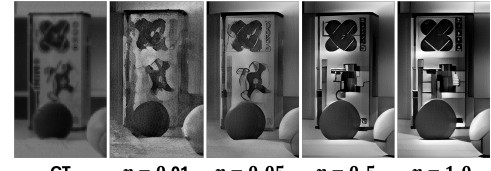

Figure 2: Comparison of frames with varying $\eta$.

weaken the EMA update and diminish scene structure guidance, reducing detail clarity. Our choice of $\eta = 0.05$ strikes a balance, providing effective guidance with minimal hallucination.

## B  COMPUTATIONAL EFFICIENCY

Tab. 1 shows that our method requires memory and inference time comparable to other diffusion-based SR methods such as MGLD-VSR. However, these demands are much higher than those of non-diffusion-based methods such as E2SRI and Hyper-E2VID due to the diffusion process and larger parameter counts for diffusion models.

Table 1: Compute efficiency comparisons.

|  | Inference(s) | Parameters | Memory(GB) |
| --- | --- | --- | --- |
| InvRec | 21.7 | 10.1M | 5.1 |
| E2SRI | 0.6 | 10.1M | 5.2 |
| Hyper-E2VID | 0.1 | 10.2M | 5.6 |
| MGLD-VSR | 10.3 | 1.4B | 28.4 |
| EventSR-Zero (Ours) | 13.1 | 1.4B | 29.7 |

## C  LIMITATIONS

As a diffusion-based model, EventSR-Zero requires more memory and inference time than non-diffusion-based methods. However, it can leverage future advancements in diffusion model efficiency, such as model distillation and low-rank adaptation, to mitigate these challenges in computational efficiency.

## D  IMPLEMENTATION DETAILS

**Iterative Contrast Refinement.**  Following Friedhelm Hamann (2024), we employ the Random Single-Reference Focus Loss (RSRFL) within ICR's CMax optimization framework. Unlike earlier multi-reference methods Shiba et al. (2022), which evaluate contrast at predefined reference points, RSRFL selects reference times using a uniform random distribution. This approach ensures that IWEs remain sharp across all reference times. Our implicit MLP architecture consists of two layers with a hidden dimension of 64. The ICR optimization runs for max_refine_iter $= 250$ iterations with a learning rate of 0.01.

**Reconditioning Guidance.**  The MGLD-VSR model is configured to super-resolve batches of 5 frames across 50 diffusion steps. During each diffusion step, we perform max_align_iter $= 100$ iterations of IWE-alignment (Eqn. 9) using the Adam optimizer Kingma & Ba (2017) with a learning rate of 0.03. To ensure consistent exposure between the super-resolved (SR) and low-resolution (LR) frames generated by Hyper-E2VID, we apply adaptive instance normalization Huang & Belongie (2017) to the output frames.

# E  PSEUDOCODE

Algorithm 1 and Algorithm 2 shows the pseudocode for ICR and RG, respectively.

---

**Algorithm 1** Iterative Contrast Refinement (ICR)

---

**Input**: LR flows $\mathbf{f}_{lr}$, LR events $e^{lr}$
**Output**: Refined flows $\mathbf{f}'_{hr}$, refined IWE $I_{iwe}$

1: $\Delta\mathbf{f}_\theta = 0$
2: **for** refine_iter $= [0, \text{max\_refine\_iter}]$ **do**
3:      $e^{hr} = \text{REPLICATEEVENTS}(e^{lr}, k)$
4:      $\Delta\mathbf{f}_\theta = \text{MLP}(\gamma(x), \gamma(y))$                        ▷ Eq. (7)
5:      $\mathbf{f}_{hr} = \mathbf{U}(\mathbf{f}_{lr}) + \Delta\mathbf{f}_\theta$                        ▷ Eq. (5)
6:      $I_{iwe} = \text{COMPUTEIWE}(\mathbf{f}_{hr}, e^{hr})$
7:      $L_{contrast} = \text{GRADIENTMAGNITUDE}(I_{iwe})$
8:      Take gradient descent step on $\nabla_\theta L_{contrast}$
9:      $\Delta\mathbf{f}_\theta^{old} \leftarrow \Delta\mathbf{f}_\theta$
10: **end for**
11: $\mathbf{f}'_{hr} = \mathbf{f}_{lr} + \Delta\mathbf{f}_\theta$
12: **return** $\mathbf{f}'_{hr}, I_{iwe}$

---

**Algorithm 2** Reconditioning Guidance (RG)

---

**Input**: Latent $z_0$, refined IWE $I_{iwe}$, refined flows $\mathbf{f}'_{hr}$, image conditioning $\mathbf{x}_t^{cond}$
**Output**: SR frame $\mathbf{x}_{sr}$

1: $\mathbf{z}_T \sim \mathcal{N}(0, I)$
2: $\eta = 0.05$
3: **for** $t = [T, 0]$ **do**
4:      $\boldsymbol{\epsilon}_\phi(z'_t, \mathbf{x}_t^{cond}, t) = \text{UNET}(z'_t, \mathbf{x}_t^{cond}, t)$
5:      $\hat{\mathbf{z}}_0 = \frac{1}{\sqrt{\bar{\alpha}_t}}(\mathbf{z}_t - \sqrt{1 - \bar{\alpha}_t}\boldsymbol{\epsilon}_\phi(\mathbf{z}_t, \mathbf{x}_t^{cond}, t))$      ▷ Eq. (10)
6:      $\hat{\mathbf{x}}_0 = \text{DECODEVAE}(\hat{\mathbf{z}}_0)$
7:      $\bar{\mathbf{x}}_0 = \hat{\mathbf{x}}_0$
8:      **for** $i = [0, \text{max\_align\_iter}]$ **do**
9:          $D_{x,y}\bar{\mathbf{x}}_0 = \mathbf{f}'_{hr}(\bar{\mathbf{x}}_0) - \bar{\mathbf{x}}_0$
10:         $L_{rg} = ||D_{x,y}\bar{\mathbf{x}}_0 - I_{iwe}||_2 + \lambda\mathcal{R}(\bar{\mathbf{x}}_0)$      ▷ Eq. (9)
11:         Take gradient descent step on $\nabla_\theta L_{rg}$
12:         $\bar{\mathbf{x}}_0^{old} \leftarrow \bar{\mathbf{x}}_0$
13:      **end for**
14:      $\mathbf{x}_t^{cond} = (1 - \eta)\mathbf{x}_t^{cond} + \eta\bar{\mathbf{x}}_0$      ▷ Eq. (11)
15: **end for**
16: $\mathbf{x}_{sr} = \text{DECODEVAE}(\mathbf{x}_0)$
17: **return** $\mathbf{x}_{sr}$

---