# OpenReview forum: "EventSR-Zero: Training-free Event Video Super-Resolution with Diffusion Priors"
_ICLR.cc/2026/Conference — Submitted to ICLR 2026_

### Official Review · Reviewer_5x4f · 2025-10-27

**Soundness:** 2
**Presentation:** 3
**Contribution:** 2
**Rating:** 4
**Confidence:** 5

**Summary:**

This paper addresses the task of Event-to-Video Super-Resolution (EVSR), aiming to reconstruct high-resolution (HR) video frames from low-resolution (LR) event streams. The authors propose a novel training-free EVSR framework that consists of an Implicit Contrast Refinement (ICR) module to extract sub-pixel scene details from LR events, which are then injected into a Reconditioning Guidance (RG) module to steer the diffusion-based video super-resolution (VSR) process. Experimental results demonstrate the effectiveness of the proposed method.

**Strengths:**

- The idea of leveraging an IWE-inspired approach to extract high-frequency details, combined with an MLP-based frequency-constrained mechanism to prevent collapse, is interesting and well-motivated.

- The integration of extracted event details to guide the diffusion-based VSR process is conceptually sound.

- The entire pipeline is training-free, which enhances its practicality and ease of deployment.

**Weaknesses:**

- **Lack of Important Baselines**: The paper defines EVSR as generating HR frames from LR events. A natural and intuitive baseline would be a two-stage approach: Event Super-Resolution (Event-SR) followed by Event-to-Video (E2V) reconstruction (e.g., EventZoom + E2VID). While the authors argue such baselines are outdated, more recent and advanced methods in both Event-SR and E2V reconstruction exist and should be included for a fair and comprehensive comparison.
- **Insufficient Justification and Ablation Studies**: The paper employs HyperE2VID for frame reconstruction and MGLD for diffusion-based VSR. However, the rationale for selecting these specific models is not clearly explained. To strengthen the validity of the approach, the authors should provide ablation studies or comparisons with other state-of-the-art E2V and diffusion-based VSR methods to demonstrate the robustness and generalizability of their design choices.
- **Concerns Regarding Effectiveness in Training-Free Setting**: The claim of being training-free is a key advantage, but also raises concerns about generalization. The pretrained MGLD model was trained on high-quality datasets, while the test data (e.g., HQF) has significantly lower quality and resolution. Without fine-tuning or adaptation, it is questionable whether the diffusion prior remains effective under such domain shift. The authors should provide more analysis or discussion on how the method handles such distribution mismatches.
- **Inappropriate Dataset Choice**: The use of outdated and low-resolution datasets (ECD and HQF, both 180×240) is problematic. Given that the method aims to recover high-frequency details, evaluating on datasets with poor spatial resolution and limited ground-truth detail undermines the credibility of the claimed improvements. The authors are encouraged to evaluate on high-quality event datasets, such as those captured with Prophesee Gen4 sensors (e.g., 1280×720), including publicly available benchmarks like BS-ERGB, to better validate the effectiveness of their approach.

**Questions:**

See weaknesses

---

### Official Review · Reviewer_jgAV · 2025-10-30

**Soundness:** 3
**Presentation:** 2
**Contribution:** 2
**Rating:** 4
**Confidence:** 4

**Summary:**

The paper proposes a training-free pipeline for event-based super-resolution. To achieve the goal, authors propose Implicit Contrast Refinement (ICR) and a Reconditioning Guidance (RG). ICR replicates low-resolution events onto a high-resolution grid with sub-pixel offsets, then refines structure in HR space by maximizing contrast while regularizing with a low-frequency positional-encoding MLP to mitigate event collapse. Second, RG step drives a diffusion-based VSR model by aligning the latent reconstruction’s gradients with the IWE and updating the conditioning image via an EMA rule for stability. To demonstrate its effectiveness, they evaluate the method on two datasets and present the results.

**Strengths:**

Motivated by a training-free paradigm, the paper tackles video super-resolution through a concise, optimization-based approach.
Across two benchmark datasets, the authors report state-of-the-art performance among the evaluated methods.
The study shows that, compared with conventional self-guidance, directly updating the conditioning image via an exponential moving average is particularly effective in the SR setting.

**Weaknesses:**

Most of the pipeline is prior research; the core contribution reads as a combination of ICR with an EMA-guided conditioning tweak rather than a new model class.
Absolute SSIM/LPIPS/MSE gains over baselines are small, and no statistical significance or per-sequence analyses are provided. The reliability of the reported metrics is questionable, as the evaluation pipeline includes downsampling (and additional preprocessing), which can bias full-reference scores.
The reconditioning step adds heavy inner optimization per diffusion step.
Experiments focus on DAVIS240-based datasets (ECD/HQF), with little coverage of higher-resolution sensors or diverse scenes.

**Questions:**

If one first does super-resolution (e.g., [1]) the events and then applies E2VID, similar results would likely be obtained; why was this reversed ordering not included as a comparison?
Experiments were conducted only at the very small resolution of 240×180; does the method perform equally well on newer, higher-resolution datasets such as CED?
Because the outputs are downsampled to compute full-reference metrics, I question how meaningful those metrics are.
Using an RGB video super-resolution dataset with LR–HR pairs, one could synthesize events with a simulator such as ESIM[2], apply the proposed method, and then compare against the HR ground truth with full-reference metrics to verify whether the improvements hold end-to-end.

[1] Weng, Wenming, Yueyi Zhang, and Zhiwei Xiong. "Boosting event stream super-resolution with a recurrent neural network." European Conference on Computer Vision. Cham: Springer Nature Switzerland, 2022.
[2] Rebecq, Henri, Daniel Gehrig, and Davide Scaramuzza. "Esim: an open event camera simulator." Conference on robot learning. PMLR, 2018.

---

### Official Review · Reviewer_8oqx · 2025-10-31

**Soundness:** 2
**Presentation:** 2
**Contribution:** 2
**Rating:** 6
**Confidence:** 3

**Summary:**

This paper deals with event-to-video super-resolution (EVSR) that reconstructs high-resolution video from low-resolution events. In details, it first uses HyperE2VID to generate the LR video and an event-based flow estimator to predict the LR flows from event data. Then, after refining the LR video with flows, it uses the MGLD-VSR for video super-resolution. The key contributions of this paper lies in the refinement of the LR video guidance (more specifically, the LR flows) for better VSR.

**Strengths:**

1, It proposes a training-free method that generates super-resolved images from LR events.

2, It proposes the implicit contrast refinement in the high-resolution domain to deal with the event collapse problem. A frequency-constrained implicit function regularizer (i.e., a MLP) is used to stabilize the optimization of LR flows.

3, It proposes a diffusion guidance stategy that transfers the fine event structures into the conditioning in diffusion-based VSR (rather than on the intermediate latents), by aligning the condition to the refined IWE.

**Weaknesses:**

1, Running the contrast maximimation in the HR space might lead to significant computation burden. Can you provide the comparison in this aspect?

2, In L320, is it reasonable to approximiate the ground-truth IWE with the spatial derivative of the intensity frame (Eq. 8)? In particular, at the beginning of diffusion, $x_0$ is noisy and is far away from $I_{gt}$.

3, Why does the IWE-aligned reconstructions lead to errors (L356)? It is the target/optimized conditioning image and should be a good LR guidance for video SR.

4, Is there a way to directly evaluate the quality of the refined flows, as well as the refined IWE?

**Questions:**

See the weakness.

---

### Official Review · Reviewer_PRb3 · 2025-10-31

**Soundness:** 3
**Presentation:** 2
**Contribution:** 2
**Rating:** 4
**Confidence:** 4

**Summary:**

This work introduces a training-free method for event-based super-resolution that integrates two key modules: Implicit Contrast Refinement (ICR) and Reconditioning Guidance (RG). The ICR module focuses on extracting sub-pixel details from low-resolution event data, while RG employs a diffusion-based video super-resolution framework to progressively enhance the conditioning frames. Experimental evaluations confirm the effectiveness of the proposed method in restoring shapes and improving visual fidelity, and ablation analyses further highlight the individual impact of each component.

**Strengths:**

The proposed training-free framework for enhancing event-based reconstruction quality is both innovative and of significant value to downstream applications. The introduced ICR and RG modules demonstrate clear novelty and are thoughtfully and rigorously designed. The manuscript is well-organized and generally easy to follow, with a clear and professional writing style. The experimental results are visually compelling, and the quantitative evaluations substantiate the method’s superior performance over state-of-the-art approaches.

**Weaknesses:**

1. Outdated Evaluation Datasets: The evaluation datasets used in the paper are relatively outdated (ECD from 2017 and HQF from 2020). Upon visual inspection, some ground truth frames shown in Figures 6 and 7 exhibit noticeable artifacts and low-quality regions, which raises concerns about the reliability of these datasets for accurate performance evaluation. In recent years, several newer benchmark datasets with higher resolution and improved fidelity have been introduced, such as EventAid [1]. The authors are encouraged to validate their method on these newer datasets to ensure a fair and up-to-date evaluation.
2. Lack of Video Results: Although the authors position their work as an event-based reconstruction approach, the paper does not provide sufficient video-level results—neither in terms of quantitative metrics nor qualitative visualizations—in the main submission. Including comprehensive video results would significantly strengthen the paper and provide clearer evidence of the proposed method’s effectiveness in dynamic scenarios.
3. Missing Analysis of Computational Efficiency: The paper currently lacks a comparison of computational efficiency, such as runtime or model complexity, between the proposed method and competing approaches. Incorporating such discussions and measurements would offer a more complete understanding of the method’s practical value and trade-offs.
4. Clarification on Novelty and Relation to Prior Work: The proposed method appears to build upon existing works such as IWE-based formulations and the approach by Zhang et al. [2], which also employ self-supervised strategies for event-based reconstruction. While the introduction of a diffusion model is a notable addition, the manuscript does not clearly delineate the methodological differences—particularly in the mathematical formulations or optimization strategies—between this work and prior studies. A more explicit discussion is recommended to better highlight the novelty and distinct contributions of the proposed approach.

[1] Duan et al. EventAid: Benchmarking Event-Aided Image/Video Enhancement Algorithms With Real-Captured Hybrid Dataset. IEEE TPAMI, 2025.
[2] Zhang et al. Formulating Event-Based Image Reconstruction as a Linear Inverse Problem with Deep Regularization Using Optical Flow. IEEE TPAMI, 2023.

**Questions:**

Please refer to Weaknesses.

---

### Official Review · Reviewer_UE2u · 2025-11-03

**Soundness:** 2
**Presentation:** 3
**Contribution:** 3
**Rating:** 2
**Confidence:** 4

**Summary:**

EventSR-Zero, a training-free framework that exploits the high temporal resolution of event cameras to recover fine-grained details from low-resolution events and uses them to guide a diffusion-based Video Super-Resolution (VSR) model in generating high-quality super-resolved videos of the underlying scene.

**Strengths:**

1. novelty
- the Implicit Contrast Refinement (ICR) and  Reconditioning Guidance (RG) seem novel.

2. performance
- seems good.
- ourperforms existing methods in qualitative comparisons.

3. presentation
- well-structured.
- Figures look appealing.

**Weaknesses:**

1. experiment
- the datasets used are very limited. only two small datasets with poor resolutions of 240 × 180 and low frame rate of 22Hz are adopted.
- experiments on some more challenging datasets (high resolution, fast motion) are missing.

2. evaluation
- since there is no real high-resolution ground truth, the evaluation still relies on downsampled comparisons. this weakens the persuasiveness of the objective metrics.
- no-reference metrics such as sim-to-real consistency or high-frame-rate optical flow consistency should be included.

3. efficiency
- lack of discussion on efficiency.
-  also, it does not provide a detailed discussion of the computational cost of the ICR and RG modules.
- since each diffusion step involves EMA updates and 100 iterations of gradient alignment optimization, the inference time may be relatively long.

**Questions:**

1. The experiments are conducted only on two relatively small datasets (ECD and HQF) with low resolutions (240 × 180) and low frame rates (22 Hz). Could the authors evaluate their method on more challenging datasets with higher resolutions or faster motion?

2. Since no real high-resolution ground truth is available, the evaluation currently relies on downsampled comparisons, which may limit the persuasiveness of the results. Have the authors considered incorporating no-reference metrics such as sim-to-real consistency or high-frame-rate optical flow consistency to provide a more comprehensive assessment?

3. The paper lacks a detailed discussion of the computational efficiency of the proposed framework. Could the authors provide runtime statistics or complexity analysis for both the ICR and RG modules?

4. Given that each diffusion step includes EMA updates and around 100 iterations of gradient alignment optimization, how long does the full inference process take? Are there potential acceleration strategies (e.g., fewer iterations, parallelization, or pruning) to make the method more practical for real-time or large-scale applications?

---

### Meta-Review · Area_Chair_vDpB · 2026-01-07

**Summary:**

This work got 1 reject, 3 marginally below, 1 marginally above ratings. There are substantial major concerns raised by the reviewers but there is no author rebuttal submitted. ACs recommend rejection.

**Reviewer Concerns:**

The same as above.

**Reviewer Scores:**

This work got 1 reject, 3 marginally below, 1 marginally above ratings, and there is no author rebuttal.

---

### Decision · Program_Chairs · 2026-01-26

Reject